# The role of intraoperative microelectrode recording and stimulation in subthalamic lead placement for Parkinson's disease

**Vesna Malinova[1]\*, Anabel Pinter[2], Cristina Dragaescu[3], Veit Rohde[1], Claudia Trenkwalder[1,3], Friederike Sixel-Döring[3☯], Kajetan L. von Eckardstein[1,4☯]**

**1** Department of Neurosurgery, Göttingen University Medical Center, Göttingen, Germany, **2** Department of Neurology, Göttingen University Medical Center, Göttingen, Germany, **3** Paracelsus Elena Clinic, Kassel, Germany, **4** Department of Neurosurgery, Westpfalz-Klinikum Kaiserslautern, Kaiserslautern, Germany

☯ These authors contributed equally to this work.

\* vesna.malinova@gmail.com

**Data Availability Statement:** All relevant data are within the paper.

**Funding:** This study was not funded.

## Abstract

### Objective

Intraoperative microelectrode recording (MER) and test-stimulation are regarded as the gold standard for proper placement of subthalamic (STN) deep brain stimulation (DBS) electrodes in Parkinson's disease (PD), requiring the patient to be awake during the procedure. In accordance with good clinical practice, most attending neurologists will request the clinically most efficacious trajectory for definite lead placement. However, the necessity of microelectrode-test-stimulation is disputed, as it may limit the access to DBS therapy, excluding those not willing or incapable of undergoing awake surgery.

### Methods

We retrospectively analyzed the MERs and microelectrode-test-stimulation results with regard to the decision on definite lead placement and clinical outcome in a cohort of 67 PD-patients with STN-DBS. All patients received bilateral quadripolar ring electrodes. To ascertain overall procedural efficacy, we calculated the surgical index (SI) by comparing preoperative motor improvement induced by levodopa to that induced by stimulation 7 to 18 months after surgery, measured as the relative difference between ON and OFF-states on the Unified Parkinson's Disease Rating Scale motor part (UPDRS-3). Additionally, a side-specific surgical index (SSSI) was calculated using the unilateral assessable items of the UPDRS-3. The SSSI where microelectrode-test-stimulation overruled MER were compared to those where the result of microelectrode-test-stimulation was congruent to MER results.

### Results

A total of 134 electrodes were analyzed. For final lead placement, the central trajectory was chosen in 54% of patient hemispheres. The mean SI was 0.99 (± 0.24). SSSI averaged 1.04 (± 0.45). In 37 lead placements, microelectrode-test-stimulation overruled MER in the final trajectory selection, in 27 of these lead placements adverse effects during microelectrode-

**Competing interests:** The authors have declared that no competing interests exist.

test-stimulation were decisive. Neither the number of test electrodes used nor the STN-signal length had an impact on the SSSI. The SSSI did not differ between lead placements with MER/microelectrode-test-stimulation congruency and those where the results of microelectrode-test-stimulation initiated lead placement in a trajectory with shorter STN signal.

## Conclusion

Intraoperative testing is mandatory to ensure an optimal motor outcome of STN DBS in PD-patients when using quadripolar ring electrodes. However, we also demonstrated that neither the length of the STN-signal on MER nor the number of test electrodes influenced the motor outcome.

## Introduction

Parkinson's disease (PD) is a progressive neurodegenerative disorder of the nigrostriatal dopaminergic system. It affects 100 to 200/100 000 patients and often manifests in the sixth decade of life. In the early stages mainly motor symptoms are present, while in later stages PD also leads to cognitive decline [1]. Pharmacotherapy with levodopa is the treatment of choice in the early stages as it adequately controls symptoms. However, in the later stages, a continuous increase of medication dosage is necessary to maintain the treatment effect, eventually leading to uncontrollable motor fluctuations and dyskinesias. High-frequency deep brain stimulation (DBS) of the subthalamic nucleus (STN) through stereotactically implanted electrodes has been shown to improve these motor symptoms [2–4]. In these studies, electrodes are placed using a combination of indirect atlas-based anatomical coordinates and electrophysiological mapping of the STN including microelectrode-test-stimulation following the initial Grenoble protocol [5]. There is, however, no consensus on the operative strategy [6–8]. Some authors dispute the necessity of intraoperative neurological evaluation of benefits and side effects of stimulation and favor surgery under general anesthesia [9, 10]. Others have shown an association between the results of intraoperative microelectrode-test-stimulation and the occurrence of motor side effects due to the unintended stimulation of the pyramidal tract [7].

To shed further light on this important question we aimed to evaluate the impact of intraoperative microelectrode recording (MER) and microelectrode-test-stimulation on intraoperative trajectory choice for permanent lead placement and on postoperative motor performance as assessed by the Unified Parkinson's Disease Rating Scale motor part (UPDRS-3) in a large patient cohort [11]. We specifically asked whether a higher number of test electrodes as a surrogate parameter for a more thorough intraoperative electrophysiological/clinical testing and a longer STN-signal on MER resulted in a better motor outcome. Furthermore, we looked for differences in motor outcome between patients with congruent microelectrode-test-stimulation and MER findings and those with predominant decision-making according to microelectrode-test-stimulation and less to MER results (i.e. final leads implanted into trajectories with an inferior STN-signal). Significant differences in motor outcome in favor of intraoperative microelectrode-test-stimulation and/or MER would indicate a possible superiority of awake compared to asleep procedure for DBS-electrode implantation.

## Methods

### Patient selection and data collection

Patients to be included in this study were identified by a prospectively maintained database of DBS patients. Only patients with idiopathic PD, who underwent STN-DBS-surgery carried out

as an awake procedure and allowing intraoperative neurological evaluation were selected for inclusion into the study. Complete documentation of intraoperative parameters was another inclusion requisite. We retrospectively reviewed intraoperative stimulation protocols. These protocols included (a) test trajectories used for MER per patient side, (b) length of typical STN-signal recorded per trajectory, (c) location of microelectrode-test-stimulation, (d) beneficial stimulation results and adverse effects per location and amplitude (voltage), as well as (e) chosen trajectory and depth of final electrode position. Furthermore, we included basic epidemiological data as well as peri- and postoperative complications.

## Indication and surgery

Indication for STN-DBS surgery was idiopathic PD with severe end-of-dose dyskinesias and significant motor symptoms. Patients were preoperatively seen by neurologists and neurosurgeons involved. As part of the preoperative assessment, an L-dopa challenge test was administered, and the results documented.

Surgery was performed in a very uniform and standardized way by three different surgeons from one center. Intraoperative electrophysiological and neurological monitoring was performed by two different neurologists. A magnetic resonance imaging (MRI) dataset according to a predefined DBS-protocol (T1 with gadolinium, T2, and susceptibility-weighted imaging) was obtained on a 3T-MRI-scanner (Magnetom® Siemens) two days prior to surgery under general anesthesia. Antidopaminergic medication was held for the operative day and patients were operated on awake under local anesthetic without sedation. After the Leksell® G-frame (Elekta, Stockholm) was placed on the regular ward, patients underwent a frame-based head CT scan with contrast medium application. The registration CT was matched with the planning MRI on either the FrameLink® or the Stealth® working station (Medtronic, Minneapolis, MN), the anterior and posterior commissure (ACPC) were manually inserted and the STN target points were calculated (12 mm lateral, 4 mm caudal, and 4 mm posterior of the mid-commissural point). Bifrontal trajectories were established, avoiding the ventricles, the cortical sulci, and blood vessels. If at all possible, we planned for five parallel needle trajectories but took the liberty of reducing the number of trajectories in case of anatomical conflicts with vessels. After the patient was brought to the operating theater, the head was shaved, prepped, and draped and coordinates were set and double-checked. Starting on the left side, a burr hole was applied, and microelectrodes were advanced to -10 mm, using a standard microdrive (Medtronic, Minneapolis) attached to the frame. From there on, recordings were obtained in millimeter increments, beginning at -5 mm in half-millimeter increments until the STN-signal vanished in all electrodes used. The length and depth of the signal were recorded. After mapping the STN electrophysiologically, stimulation testing was performed using the trajectories with promising STN-signals. Results were documented for each microelectrode-test-stimulation including location, amplitude, and clinical results. An agreement was reached between the testing neurologist and the neurosurgeon on where to implant the final lead (quadripolar 3389 lead, Medtronic, Minneapolis), taking into consideration the clinical results of the microelectrode-test-stimulation testing as well as of the MER. The final lead was implanted and the position was confirmed using lateral X-ray. The patient underwent general anesthetic and an Activa® PC impulse generator (Medtronic, Minneapolis) was implanted in the left subclavicular region. Patients were transferred to the neurology service two weeks after the operation for testing and programming. A follow-up visit was scheduled one year after the procedure for generator adjustments. During this routine visit, we performed a medication off and stimulation on/off testing to evaluate the procedural efficacy.

## Primary outcome parameters

As a clinical outcome parameter, UPDRS-3 was used, the motor subset of the UPDRS [11]. This score evaluates 27 motor items, including gait stability and other axial motor tasks. However, tremor, rigidity, and fine motor skills are recorded side-specifically and can be evaluated as such. Preoperatively, the UPDRS-3 score was documented off medication and after L-Dopa administration (L-Dopa challenge test). Relative motor improvement was recorded as a percentage (1 –[UPDRS-3 ON medication/UPDRS-3 OFF medication]). Improvement scores were usually between 40% and 100% and were regarded as a prerequisite for surgery.

Similarly, a postoperative stimulation test was performed at one year. Relative motor improvement off medication was recorded as a percentage (1 –[UPDRS-3 ON stimulation OFF medication/UPDRS-3 OFF stimulation OFF medication]). The surgical index, a parameter measuring the procedural quality, is defined as the quotient of preoperative and postoperative testing (L-Dopa challenge test/stimulation test) and is hence dimensionless. A surgical index of <1 denotes patients in which the relative reduction of UPDRS-3 points upon stimulation exceeds the preoperative L-Dopa challenge test. To evaluate the procedural effects of planning, adjustments, and implantation of single electrodes, side-specific documentation of the procedural efficacy is necessary. In analogy to the surgical index described above, we calculated a side-specific SI based upon those UPDRS-3 items recorded side-specifically. However, due to fewer items used, the mean variation of the calculated side-specific surgical indexes increased. To enable evaluation of single electrode placement, and hence, hemispheres we extracted side-specific items of the UPDRS-3 and correlated these to intraoperative MER and microelectrode-test-stimulation results as well as to intraoperative decision-making concerning the trajectory for permanent lead placement.

## Ethical requirements

Approval for evaluation was obtained from the ethics committee of the Medical University of Göttingen (Number 21/3/19). This study was conducted according to the principles of the Helsinki Declaration [12]. A patient's consent for treatment was obtained according to the individual institutional guidelines. Due to a retrospective analysis of the data for this study, additional informed consent was deemed unnecessary.

## Statistical analysis

A big part of statistical analysis in this manuscript consists of descriptive statistics. We used t-test for the evaluation of differences between the patient group with decisions made in accordance with MER compared to the group where the decisions were made according to the test stimulations and not to the MER. For descriptive statistics and plotting, Excel spreadsheet software (Microsoft, Redmond) was used. For t-tests employed in comparing surgical indexes between different groups, $p < 0.05$ was regarded as significant.

# Results

## Study population

Between January 2010 and June 2016, a total of 137 patients underwent bilateral subthalamic lead placement for idiopathic PD in our department. Of 124 patients with full intraoperative documentation of MER and microelectrode-test-stimulation findings, 116 patients were operated awake under local anesthesia. We excluded ten patients for a variety of clinical reasons (infection requiring surgical revision and system removal within six months (n = 3), significant intracranial hemorrhage (n = 2), intraoperative re-calculation of coordinates (n = 2), diagnosis

of atypical Parkinson's syndrome/multisystem atrophy on follow-up (n = 2), previous thalamic electrode placement elsewhere (n = 1), and abortion of the procedure due to an intraoperative seizure (n = 1). Of the remaining 106 patients, 39 patients did not report back between 7 and 18 months after surgery for their routine clinical one-year follow-up examination, refused further clinical testing off-medication, or were lost to follow-up. Thus, 67 patients receiving a total of 134 electrodes were included in this retrospective analysis (mean age 60.1 years, 65.6% male).

## Surgical complications

The postoperative course was completely unremarkable in 62/67 patients (92.5%). Two patients (3%) required antibiotics for a superficial wound infection postoperatively, one patient (1.5%) underwent local wound revision surgery for dehiscence 16 months after implantation, and one patient (1.5%) required a hematoma evacuation of the pectoral wound on the evening of surgery. One patient (1.5%) required intensive care unit observation for a perioperative myocardial infarction.

## Intraoperative findings of MER and microelectrode-test-stimulation

A total of 525 microelectrodes were used, averaging 3.9 recordings per patient hemisphere; four or five microelectrodes were used in the evaluation of 94 hemispheres (Fig 1A). The average length of a typical STN signal over all microelectrodes used was 3.2 mm (+/- 2.4 mm), ranging from 0 to 8.0 mm. The average length of STN signal along the central trajectory was 4.1 mm (+/- 2.2 mm), along the anterior trajectory 4.1 mm (+/- 2.2 mm), along the lateral trajectory 3.2 mm (+/-2,4 mm), along the posterior trajectory 1.8 mm (+/- 2.0 mm), and along the medial trajectory 1.7 mm (+/-2.0 mm). Microelectrode-test-stimulation was performed in a total of 501 different coordinates (electrode positions), using increasing amplitudes, averaging at 3.7 stimulation sequences per patient hemisphere; two or more trajectories were used in the evaluation of 87 hemispheres. The final trajectory chosen for implantation of the permanent electrode was the central position in 73 patient hemispheres (54%) and in 36 hemispheres (27%) the anterior position (Fig 1B).

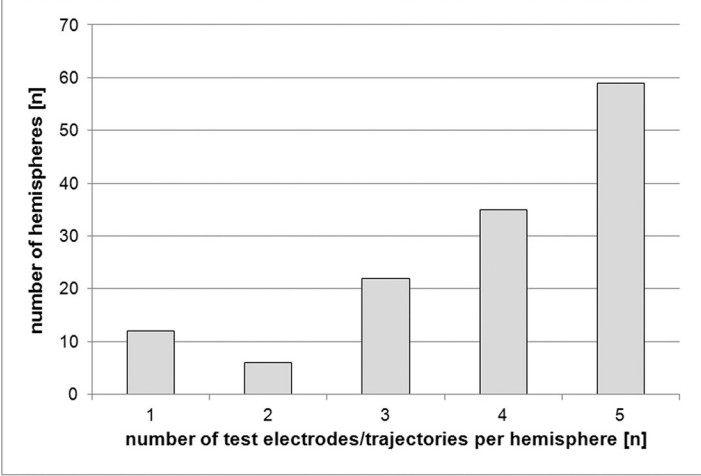 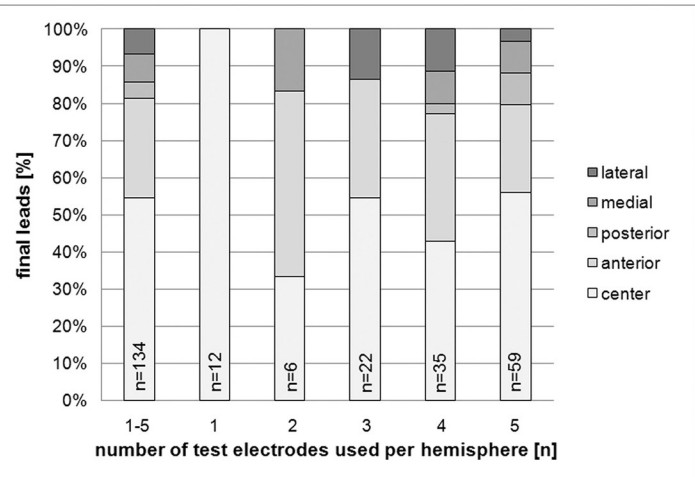

**Fig 1.** (a, left) Number of test electrodes used per patient hemisphere (525 test electrodes in total). (b, right) Final electrode location per number of microelectrodes used for recording.

In 37 patient hemispheres (27.6%), the final trajectory chosen for implantation of the permanent electrode was not the trajectory with the longest STN signal, including those recordings where two or more signals were of equal length. The reasons for changing the trajectory of permanent lead position was a clinical benefit on the cardinal motor symptoms of PD during microelectrode-test-stimulation in 10 patient hemispheres, whereby adverse effects partly or fully determined the implantation trajectory in the remaining 27 patient hemispheres (Fig 2A). We observed stimulation side effects in all but one patient but only in 27 patient hemispheres did the observed adverse effects ultimately have an impact on surgical decision-making. Those adverse effects included oculomotor symptoms in 12 patient hemispheres, motor symptoms in 11 patient hemispheres, non-transient sensory symptoms in two patient hemispheres, difficulties with speech in six patient hemispheres, and in nine patient hemispheres (33.3%) diffuse symptoms that were summarized as general discomfort or nausea (Fig 2B).

## Motor improvement in relation to intraoperative MER and microelectrode-test-stimulation findings

As expected, our patients did very well after surgery. Overall, the surgical index was 0.99, hence stimulation after one year off medication resulted in an improvement in UPDRS-3 identical to the preoperative L-Dopa test (Fig 3A). In our cohort, age was not a predictor of better surgical outcomes, however, as a result of very careful preoperative screening the oldest patients included were only 71 years old (Fig 3B). Neither the number of test electrodes used nor the maximum length of STN-signal on MER or whether microelectrode-test-stimulation findings overruled the MER results during intraoperative decision-making concerning the trajectory chosen for permanent lead placement, were associated with a better side-specific motor outcome (Fig 4). To overcome the issue of increased mean variations of the side-specific outcome parameter we also looked into patients in whom the STN signal was bilaterally uniformly short or long and correlated these two groups to the overall surgical index. However, a correlation again could not be established (Fig 5).

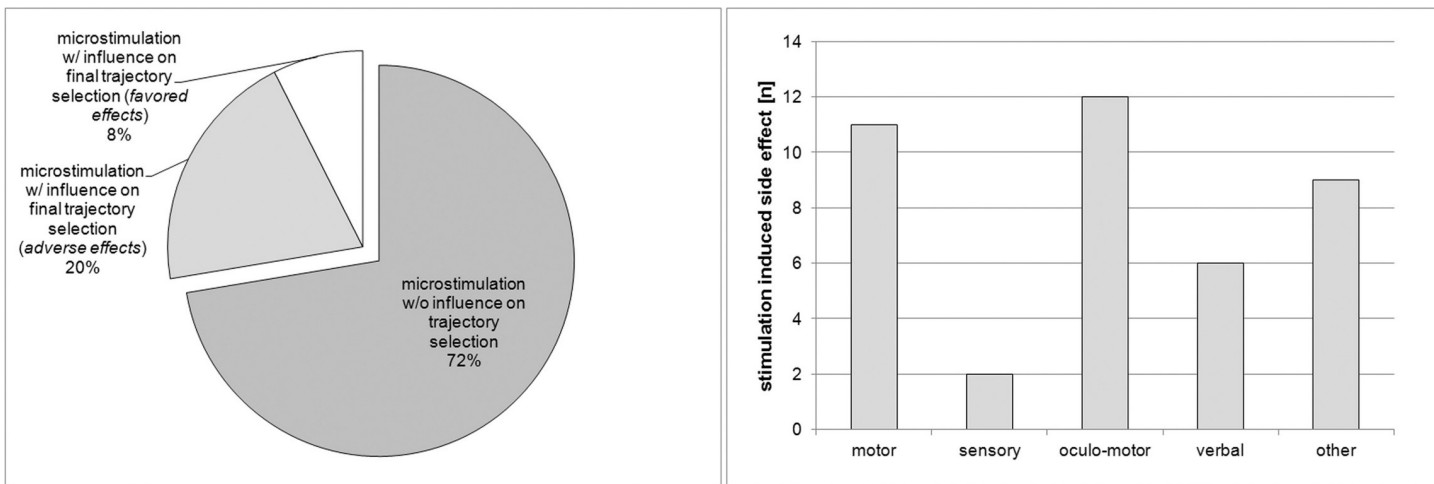

**Fig 2.** (a, left) Distribution of patient hemispheres in which microstimulation had no influence on final implantation tract selection (n = 97) and in which stimulation induced side effects (n = 27) and favored effects (n = 10) influenced final trajectory selection. (b, right) Stimulation induced side effects in 27 patient hemispheres influencing final trajectory selection. The right-hand coloum marked 'other' included patients suffering from nausea, dizziness and lightheadedness, sweating, and the feeling of inner heat, or just felt "funny".

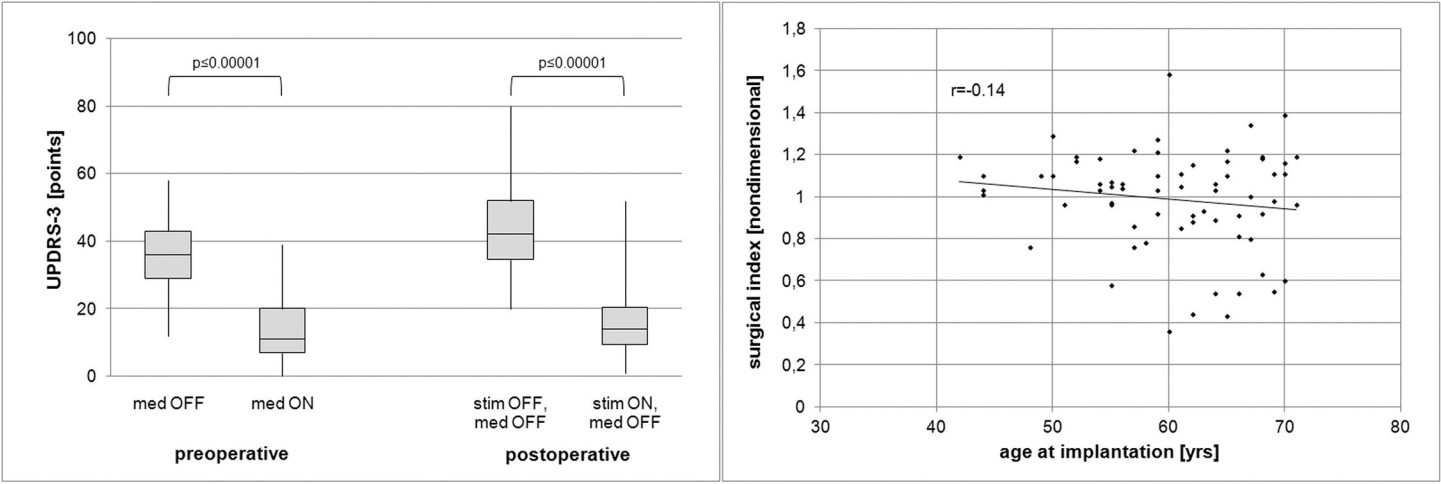

**Fig 3.** (a, left) Boxplot presentation of preoperative UPDRS-3 scores off-medication and on-medication (L-Dopa challenge test) and postoperative scores off-stimulation off-medication and on-stimulation off-medication at one year. Boxes represent first quartiles, median, and third quartiles, whiskers depict minimum and maximum scores. (b, right) Correlation of age and surgical outcome.

## Discussion

The discussion about the necessity of intraoperative clinical monitoring is not settled. Intraoperative microelectrode-test-stimulation–only possible in awake procedures–mimics the postoperative stimulation situation and should hence improve optimal lead position and therefore outcome. However, retrospective patient series comparing awake and asleep procedures, the latter without the possibility of intraoperative clinical monitoring, have failed to show a clear superiority of either procedure [13–16]. A large multicenter prospective trial randomly assigning patients to either procedure has not been planned so far. Being an advocate of performing the procedure awake with full possibilities to optimize lead positioning, we reviewed our experience concerning intraoperative monitoring in lead placement for PD in a very homogenous patient cohort and correlated the intraoperative parameters with an objective outcome

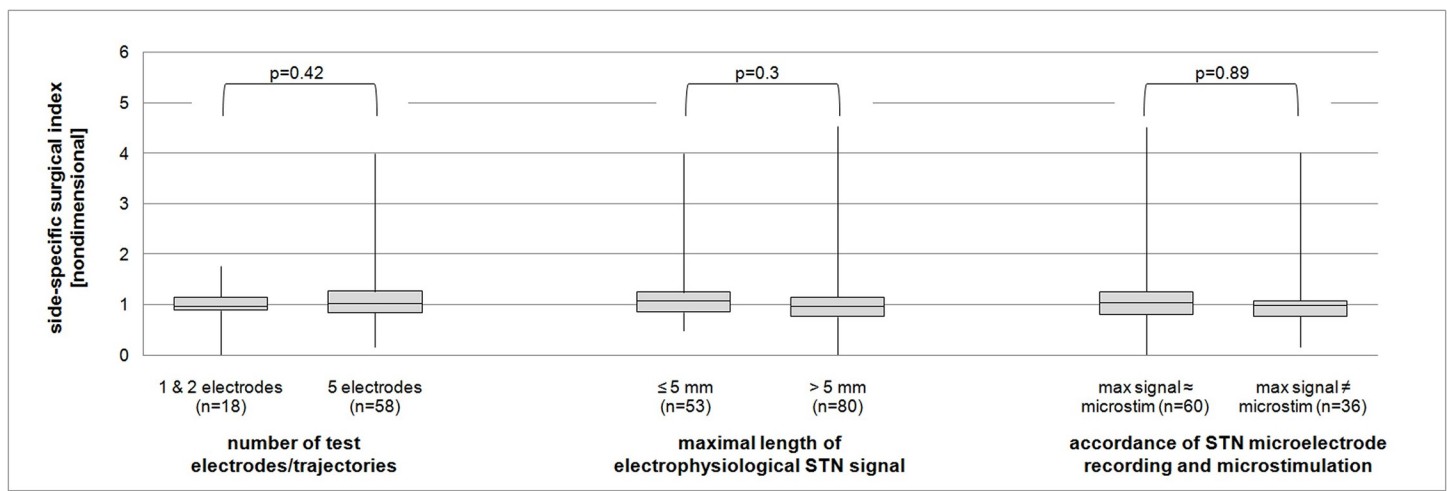

**Fig 4. Boxplot presentation of correlation of side-specific motor results with different intraoperative parameters.** Boxes represent first quartiles, median, and third quartiles, whiskers depict minimum and maximum scores. Left. Influence of number of test electrodes used on outcome. Middle. Influence of maximal length of STN-signal on outcome. Right. Influence of accordance of STN-signal and microstimulation results on outcome.

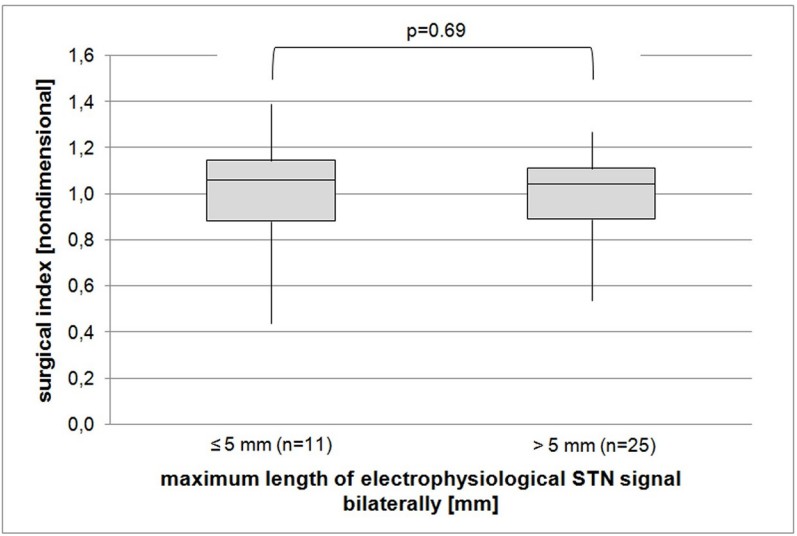

**Fig 5. Correlation of bilateral uniform electrophysiologic STN length to the operative outcome.** Boxes represent first quartiles, median, and third quartiles, whiskers depict minimum and maximum scores.

parameter. One of the problems of such an evaluation is that the independent variable (i.e. intraoperative monitoring and lead placement) is reported referring to the hemisphere, the dependent variable (i.e. outcome) is usually reported referring to the patient. We have addressed this problem by calculating a side-specific outcome parameter. We focused on motor tasks, which intraoperatively had a high impact on final tract selection, and assessed side-specific tremor, rigidity, and fine motor skills. As others have demonstrated [17], MRI-based coordinates were corrected in about 20 to 40% of hemispheres by the results of MER and intraoperative microelectrode-test-stimulation. Similarly, independent of the number of test electrodes, we placed the final electrode into the center trajectory in 54.4%.

We expected to see an improved outcome in patient hemispheres in which we were able to place five test electrodes, offering the full range of intraoperative testing. However, neither the number of test electrodes nor the length and quality of the electrophysiological STN signal correlate with a better side-specific motor outcome. Although the mean variations were high, due to the reduced number of items evaluated in a side-specific fashion and a statistical significance was therefore hard to reach, the median score also did not show a trend. Even comparing patients with strong STN signals bilaterally with those with short signal bilaterally did not show any difference in patient-specific motor outcome at one year, although the mean variation was smaller due to the full number of items assessed on the UPDRS-3. Therefore, based upon the patient cohort presented here, the motor outcome at one year was independent of the results of intraoperative MER and of microelectrode-test-stimulation results. We were not able to receive a better motor outcome in patients with extensive intraoperative monitoring. These results–unexpectedly–are a strong intercessor for less vigorous employment of awake surgeries in STN-DBS procedures for PD. According to the current literature, the role of MER for reliable neurophysiological mapping of STN, and hence higher DBS-efficacy, remains inconclusive. While measurable differences of atlas/imaging-based STN-localization and electrophysiologically determined STN targets, as well as a better clinical outcome (UPDRS) by using intraoperative MER compared to intraoperative test-stimulation alone have been reported in some studies [14, 16, 18], other previously published studies could not show a significant difference in long-term clinical outcome between awake procedures allowing MER

and intraoperative test stimulation and asleep procedures without intraoperative neurophysiological evaluation [13, 15]. Due to limitations such as retrospective study design and a small study cohort, the results of the most previously published studies do not allow reliable conclusions to be drawn for clinical practice. Very recently, Bjerknes et al. [19] compared two randomized groups of patients, using UPDRS-3 at one year. While in one group only one microelectrode was used for testing, up to five microelectrodes were allowed in the other group. Mean improvement of 26 points for the single electrode group and of 35 points for the multiple electrode group were reported. The conclusion of this carefully designed study is somewhat different from our study, however, randomization led to an unbalanced preoperative impairment score. One explanation for the still inconclusive results in the literature concerning the impact of intraoperative MER and microelectrode-test-stimulation on the motor outcome might be the variance in the time point of outcome assessment associated with varying consideration of long-term stimulation effects. Another variable that should be considered is the programming process of the stimulation parameters itself, which can individually vary and might, therefore, influence the outcome by overcoming side-effects in some patients better than in others. Future prospective studies considering these variables are necessary to overcome these issues and to allow more reliable conclusions for clinical practice. Additionally, the increasing use of directional leads is expected to have a relevant impact on the future surgical strategy during DBS-procedures as well. Last but not least, the continuous development of imaging techniques with the possibility of better STN visualization also contributes to a constant evolving in this field.

## Limitations of the study

As an objective outcome parameter, we used the UPDRS-3, which is the subset of motor items of the widely accepted UPDRS. Using 27 parameters, a total of 108 points can be calculated for a maximum degree of impediment. This enables a very subtle detection of differences in motor skills. However, when calculating side-specific impairment, we relied on only eight items with a total of 64 possible points. This narrows the detection of small differences and widens the mean variation and carries the risk of underestimating the effects on symmetrical functions such as gait and axial stability. Other symptoms of PD, which are usually not affected by subthalamic stimulation, were not assessed (UPDRS-1: cognition, UPDRS-2: activities of daily life, UPDRS-4: complications of medical therapy). We did not routinely use a quality of life score in our patients as part of the regular clinical workup. This is a retrospective evaluation of prospectively collected data. The design represents a clinical setting yet bears the risk of selection bias. Patients experiencing very poor clinical results might have deliberately missed their one-year follow-up appointment to see another team and could hence have been excluded. However, this study attempted to focus on the effects of MER and microelectrode-test-stimulation and not on the overall clinical results of all patients operated on. Due to the increasing number of directed leads implanted, intraoperative testing might become less important, as postoperative corrections of the electrical field are possible [20]. Future research with a direct comparison between concentric quadripolar leads and directional octipolar leads is warranted.

## Conclusion

As shown above, the motor outcome at one year was largely independent of intraoperative MER and microelectrode-test-stimulation results in this patient cohort. While similar studies have come to another conclusion, intraoperative testing requiring an awake procedure seems to be less important than previously thought. Through the advent of directional octipolar lead

technology facilitating better adjustments of the electrical field, intraoperative testing might only be required in particular cases, while the standard STN-DBS implantation procedure could be performed asleep.

## Acknowledgments

This study contains parts of the doctoral thesis (M.D.) of Anabel Pinter, MD.

## Author Contributions

**Conceptualization:** Friederike Sixel-Döring, Kajetan L. von Eckardstein.

**Data curation:** Anabel Pinter, Veit Rohde.

**Formal analysis:** Anabel Pinter, Friederike Sixel-Döring, Kajetan L. von Eckardstein.

**Investigation:** Vesna Malinova, Anabel Pinter.

**Methodology:** Kajetan L. von Eckardstein.

**Project administration:** Anabel Pinter, Veit Rohde.

**Resources:** Veit Rohde, Claudia Trenkwalder.

**Supervision:** Veit Rohde, Claudia Trenkwalder, Friederike Sixel-Döring.

**Validation:** Vesna Malinova, Cristina Dragaescu, Claudia Trenkwalder, Friederike Sixel-Döring.

**Writing – original draft:** Vesna Malinova, Kajetan L. von Eckardstein.

**Writing – review & editing:** Vesna Malinova, Cristina Dragaescu, Veit Rohde, Claudia Trenkwalder, Friederike Sixel-Döring, Kajetan L. von Eckardstein.

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
