## [Decision Letter · Decision Letter 0]

3 Sep 2020

PONE-D-20-20225

The role of intraoperative microelectrode recording and stimulation in subthalamic lead placement for Parkinson’s disease

PLOS ONE

Dear Dr. Vesna Malinova

Thank you for submitting your manuscript to PLOS ONE. After careful consideration, we feel that it has merit but does not fully meet PLOS ONE’s publication criteria as it currently stands. Therefore, we invite you to submit a revised version of the manuscript that addresses the points raised during the review process.

I would appreciate if you careful attention in your response to the reviewer's comments.

We look forward to receiving your revised manuscript.

Kind regards,

Ehab Farag, MD FRCA FASA

Academic Editor

PLOS ONE

Journal Requirements:

Reviewers' comments:

Reviewer's Responses to Questions

**Comments to the Author**

1. Is the manuscript technically sound, and do the data support the conclusions?

Reviewer #1: Yes

2. Has the statistical analysis been performed appropriately and rigorously? 

Reviewer #1: I Don't Know

3. Have the authors made all data underlying the findings in their manuscript fully available?

Reviewer #1: No

4. Is the manuscript presented in an intelligible fashion and written in standard English?

Reviewer #1: Yes

5. Review Comments to the Author

Reviewer #1: The findings from this report are valuable to publish especially since this technique is being used less frequently although I do not think they are breaking any new ground here. Awake versus asleep DBS surgery for unclear reasons has been pushed to the center as highly relevant to outcomes. It may be that both approaches are safe and effective.

I did not see the data sheet attached to the PDF. The authors should include this to comply with PLOS ONE.

6. PLOS authors have the option to publish the peer review history of their article (what does this mean?). If published, this will include your full peer review and any attached files.

Reviewer #1: No

---

## [Author Response · Author response to Decision Letter 0]

13 Oct 2020

Dear Editors and Reviewers,

thank you for critically reviewing our manuscript "The role of intraoperative microelectrode recording and stimulation in subthalamic lead placement for Parkinson’s disease". We are grateful for the valuable reviewer comments. We revised the manuscript under consideration of all reviewer comments and addressed all reviewers’ requests. 

Comments to the Author

1. Is the manuscript technically sound, and do the data support the conclusions?

Reviewer #1: Yes

2. Has the statistical analysis been performed appropriately and rigorously? 

Reviewer #1: I Don't Know

A big part of statistical analysis in this manuscript consists of descriptive statistics. We used t-test for the evaluation of differences between the patient group with decisions made in accordance with MER compared to the group where the decisions were made according to the test stimulations and not to the MER. We stated this in the chapter “Statistical analysis” of the methods section of the manuscript. 

3. Have the authors made all data underlying the findings in their manuscript fully available?

The PLOS Data policy requires authors to make all data underlying the findings described in their manuscript fully available without restriction, with rare exception (please refer to the Data Availability Statement in the manuscript PDF file). The data should be provided as part of the manuscript or its supporting information or deposited to a public repository. For example, in addition to summary statistics, the data points behind means, medians and variance measures should be available. If there are restrictions on publicly sharing data—e.g. participant privacy or use of data from a third party—those must be specified.

Reviewer #1: No

We provided a PDF-file including all data underlying the findings in this manuscript. 

4. Is the manuscript presented in an intelligible fashion and written in standard English?

Reviewer #1: Yes

5. Review Comments to the Author

Reviewer #1: The findings from this report are valuable to publish especially since this technique is being used less frequently although I do not think they are breaking any new ground here. Awake versus asleep DBS surgery for unclear reasons has been pushed to the center as highly relevant to outcomes. It may be that both approaches are safe and effective.

I did not see the data sheet attached to the PDF. The authors should include this to comply with PLOS ONE.

We agree with the reviewer that probably both approaches are safe and effective. However, we consider it as an important objective to evaluate the impact of intraoperative MER during awake DBS surgery on the long-term motor outcome of the patients. From patients’ perspective asleep DBS procedure is the more comfortable and therefore, preferable option, compared to awake procedure. Hence, providing evidence towards comparable results of both awake and asleep DBS surgery would facilitate the clinical decision-making in this field. 

We provided a PDF-file including all data underlying the findings in this manuscript. 

Best regards

Vesna Malinova

---

## [Editor Report · Decision Letter 1]

21 Oct 2020

The role of intraoperative microelectrode recording and stimulation in subthalamic lead placement for Parkinson’s disease

PONE-D-20-20225R1

Dear Dr. Vesna Malinova

We’re pleased to inform you that your manuscript has been judged scientifically suitable for publication and will be formally accepted for publication once it meets all outstanding technical requirements.

Kind regards,

Ehab Farag, MD FRCA FASA

Academic Editor

PLOS ONE
---

## [Editor Report · Acceptance letter]

26 Oct 2020

PONE-D-20-20225R1 

The role of intraoperative microelectrode recording and stimulation in subthalamic lead placement for Parkinson’s disease 

Dear Dr. Malinova:

I'm pleased to inform you that your manuscript has been deemed suitable for publication in PLOS ONE. Congratulations! Your manuscript is now with our production department. 

Kind regards, 

on behalf of

Dr. Ehab Farag 

Academic Editor

PLOS ONE